# Comparison of Circulating Tumour DNA and Extracellular Vesicle DNA by Low-Pass Whole-Genome Sequencing Reveals Molecular Drivers of Disease in a Breast Cancer Patient

**DOI:** 10.3390/biomedicines9010014

**Published:** 2020-12-25

**Authors:** Olivia Ruhen, Bob Mirzai, Michael E. Clark, Bella Nguyen, Carlos Salomon, Wendy Erber, Katie Meehan

**Affiliations:** 1Translational Cancer Pathology Laboratory, School of Biomedical Sciences, The University of Western Australia, Perth 6009, Australia; olivia.ruhen@icr.ac.uk (O.R.); bob.mirzai@uwa.edu.au (B.M.); bella.nguyen@research.uwa.edu.au (B.N.); wendy.erber@uwa.edu.au (W.E.); 2Sarcoma Molecular Pathology Team, The Institute of Cancer Research, Sutton SM2 5NG, UK; 3PathWest Laboratory Medicine, Perth 6009, Australia; 4Melanoma Research Group, School of Medical and Health Sciences, Edith Cowan University, Perth 6027, Australia; m.clark@ecu.edu.au; 5Department of Medical Oncology, Sir Charles Gairdner Hospital, Perth 6009, Australia; 6Exosome Biology Laboratory, Centre for Clinical Diagnostics, UQ Centre for Clinical Research, Royal Brisbane and Women’s Hospital, The University of Queensland, Brisbane 4029, Australia; c.salomongallo@uq.edu.au; 7Department of Pharmacology, Faculty of Biological Sciences, University of Concepcion, Concepción 4030000, Chile; 8Department of Otorhinolaryngology, Head and Neck Surgery, Chinese University of Hong Kong, Hong Kong 999077, China

**Keywords:** breast cancer, biomarkers, extracellular vesicle DNA, circulating tumour DNA, liquid biopsy

## Abstract

There is increasing recognition of circulating tumour DNA (ctDNA) as a non-invasive alternative to tumour tissue for the molecular characterisation and monitoring of disease. Recent evidence suggests that cancer-associated changes can also be detected in the DNA contained within extracellular vesicles (EVs). As yet, there has been limited investigation into the relationship between EV DNA and ctDNA, and no studies have examined the EV DNA of breast cancer patients. The aim of this study was to use low-pass whole-genome sequencing to identify copy number variants (CNVs) in serial samples of both ctDNA and EV DNA from a patient with breast cancer. Of the 52 CNVs identified in tumour DNA, 36 (69%) were detected in at least one ctDNA sample and 13 (25%) in at least one EV DNA sample. The number of detectable variants in ctDNA and EV DNA increased over the natural history of the patient’s disease, which was associated with progression to cerebral metastases. This case study demonstrates that, while CNVs are detectable in patient EV DNA, ctDNA has greater sensitivity than EV DNA for serial monitoring of breast cancer.

## 1. Introduction

Copy number variants (CNVs) are among the most prevalent genomic alterations in breast cancer, with certain CNVs such as the amplification of *ERBB2* or the loss of heterozygosity at the *BRCA1/2* loci associated with particular clinical phenotypes [1,2,3]. As such, the detection of clinically relevant CNVs is of considerable prognostic and therapeutic importance in breast cancer. Whole-genome sequencing (WGS) of patient tumours has facilitated the discovery of a huge array of CNVs across the breast cancer genome; however, molecular profiling of tumour tissue is limited by sampling bias, tumour heterogeneity and lack of sample availability upon the emergence of metastatic lesions [1,4]. Accordingly, there is a need for a method that can encapsulate tumoural heterogeneity and monitor tumour clonal dynamics throughout the course of a patient’s disease.

Circulating tumour DNA (ctDNA), small fragments of DNA that have been released into the circulation by apoptotic tumour cells, represents a potential surrogate for tumour tissue that can be sampled non-invasively to track clonal evolution and therapeutic response [4]. Low-pass WGS (LP-WGS) studies of ctDNA in breast cancer have determined that CNVs associated with metastatic progression and treatment response can be observed longitudinally throughout patient treatment [5,6]. Recent data have also indicated that cancer-related CNVs can be detected in the DNA of cancer patient-derived extracellular vesicles (EVs), which are membrane-bound vesicles released from a wide range of cells including tumour cells via exocytosis or membrane budding [7,8,9,10,11,12]. Interestingly, EVs are thought to be stable in the blood for up to 30 days, and the DNA within them is protected from nuclease degradation by the vesicle lipid membrane, suggesting that the assessment of EV DNA may be a promising alternative or complement to ctDNA for non-invasive cancer diagnosis and monitoring [13,14,15]. However, the association between ctDNA and EV DNA and the degree to which their copy number profiles reflect one another remains unclear. Additionally, there have been no studies investigating the utility of EV DNA for copy number profiling in breast cancer patients.

In this case study, we used LP-WGS to identify and compare CNVs in tumour tissue, ctDNA and EV DNA from a patient with metastatic breast cancer. We hypothesised that EV DNA contains CNVs present in the patient’s tumour and also reflects those in patient ctDNA, and hence may be a potential source of breast cancer DNA for non-invasive patient monitoring.

## 2. Experimental Section

### 2.1. Patient Samples

The work presented in this study was undertaken using matched samples of tumour tissue and plasma from a patient with Stage IV breast cancer. The patient gave informed written consent prior to participation in the study. Serial blood samples (×3) were collected at routine clinical consultations in March, May and September 2016. Blood samples (10–20 mL) were collected in ethylenediaminetetraacetic acid (EDTA) anticoagulant and processed within 2 h of collection. The blood was separated by centrifugation at 1600× *g* for 20 min and the plasma fraction clarified by centrifugation at 16,000× *g* for 10 min. The supernatant was stored at −80 °C until further use. Formalin-fixed paraffin-embedded (FFPE) tumour tissue from surgical resection of tumours in April 2010, June 2014 and December 2016 was obtained from the pathology archives at PathWest Laboratory Medicine, Western Australia. All FFPE specimens were confirmed by an expert breast pathologist to possess a tumour cell content ≥ 80%.

### 2.2. EV Isolation

EVs were isolated from plasma using ultracentrifugation. Plasma samples (6 mL) were diluted 1/10 in phosphate-buffered saline (PBS) and centrifuged at 12,000× *g* for 45 min in the Optima L-90K Ultracentrifuge (Beckman Coulter, Lane Cove, Australia) to pellet any cell debris and apoptotic bodies. The resulting supernatant was centrifuged twice at 110,000× *g* for 90 min to pellet the EVs, which were resuspended in 100 μL PBS. Validation of successful EV isolation using this approach was performed on representative samples according to guidelines established by the International Society of Extracellular Vesicles [16]. This involved the use of 3 distinct techniques to elucidate particle size, yield and presence of common markers. A detailed methodology for each technique is outlined below.

### 2.3. Immuno-Electron Microscopy

EVs resuspended in PBS (10 μL) were fixed in 2% paraformaldehyde and transferred onto 200-mesh formvar–carbon-coated copper grids (ProSciTech, Kirwan, Australia). Samples were adsorbed for 20 min at room temperature prior to being washed twice in filtered PBS and 4 times in 50 mM glycine to quench free aldehyde groups. Samples were then blocked in 5% bovine serum albumin (BSA) for 10 min before being incubated with primary antibody solution (mouse anti-human CD9, clone MM2/57; 10 μg/mL, Merck, Bayswater, Australia) for 30 min. Grids were then washed 4 times in 0.1% BSA and 4 times in 0.5% BSA before incubation with secondary antibody (goat anti-mouse IgG-gold conjugate, 1:24, Aurion, Wageningen, The Netherlands) for 20 min. Labelled samples were washed 6 times in PBS, fixed in 1% glutaraldehyde and washed 6 times in deionised water before counterstaining with 1% uranyl acetate. Following 2 min of counterstaining, grids were left overnight to dry and then were visualised on the JEOL JEM-2100 electron microscope (JEOL, Frenchs Forest, Australia) at an operating voltage of 120 kV. Images were captured using an 11-megapixel Gatan Orius digital camera (Gatan, Abingdon, UK).

### 2.4. Nanoparticle Tracking Analysis

EV samples (100 μL) were diluted 1/100 in PBS and injected into the analysis chamber of a NanoSight NS500 Instrument (Malvern Panalytical, Taren Point, Australia) for particle size and concentration analysis. This instrument is equipped with a 405 nm laser and a sCMOS camera to detect the Brownian motion of light-scattering particles as they move through solution. Sample analysis was performed at a camera level of 10 and a gain of 250, with a detection threshold of 10 pixels. Settings for blur, minimum track length and minimum expected size were set to ‘auto’. Videos were recorded for 60 s at 30 frames/second in triplicate at 25 °C. All post-acquisition settings remained constant between samples. NTA software v3.0 (Malvern Panalytical, Taren Point, Australia) was used to process and analyse the data. Each video was analysed to generate particle size (nm) distribution profiles and concentration values (particles/mL solution), which were downloaded as a report with the results of the quality control analysis. The raw observational data were exported into Microsoft^®^ Excel (Microsoft, Redmond, WA, USA) as comma-separated values (CSV) file.

### 2.5. Western Blot

EV samples and MDA-MB-231 breast cancer cells in 100 μL PBS were mixed with 100 μL of RIPA buffer (Sigma-Aldrich, North Ryde, Australia) and protease inhibitors (Roche Diagnostics, North Ryde, Australia) and incubated on ice for 1 h to lyse the vesicle membranes. EV and cell proteins were collected by centrifugation at 16,000× *g* for 15 min at 4 °C. The proteins were diluted in 4× Laemmli Buffer (Bio-Rad, Gladesville, Australia) for a final loading volume of 20 μL. Proteins were separated at 155 V for 30 min on a Mini Protean^®^ TGX 5–15% Stain-Free™ Precast Gel (Bio-Rad) using the Mini Protean^®^ Tetra Cell (Bio-Rad). Proteins were then transferred onto a Trans-Blot^®^ Turbo™ Mini Nitrocellulose Membrane (Bio-Rad) using the Trans-Blot^®^ Turbo™ Transfer System (Bio-Rad) at a constant voltage of 25 V. The membrane was then probed for EV proteins using the iBind™ Flex Western Device (Thermo Fisher Scientific, Scoresby, Australia), commencing with blocking for 20 min in iBind™ Flex Solution (Thermo Fisher Scientific). Primary antibody solutions of mouse anti-human CD9 (clone 1.3.3.22, Thermo Fisher Scientific), mouse anti-human calnexin (clone 1563, Novus Biologicals, Littleton, CO, USA), and rabbit anti-human TSG101 (clone EPR7130B, Abcam, Melbourne, Australia) were created, with dilutions of 1:250 and 1:500 for anti-CD9 and calnexin/TSG101, respectively. Samples were incubated with primary antibody solution before being incubated with secondary antibodies (sheep anti-mouse IgG-HRP conjugate, 1:2000, GE Healthcare, Parramatta, Australia; goat anti-rabbit IgG-HRP conjugate, 1:2000, Merck) for a combined run time of 2.5 h. Signals were developed using Clarity™ Western ECL Blotting Substrates (Bio-Rad) and were subsequently imaged using the ChemiDoc™ Touch Imaging System (Bio-Rad). Images were processed using Image Lab™ software v6.0 (Bio-Rad, Gladesville, Australia).

### 2.6. DNA Extraction

EVs (50 μL) were treated with DNase I prior to DNA extraction to reduce the levels of cell-free DNA, which was co-isolated along with the EVs. This was carried out with the Ambion TURBO DNA-free DNA Removal Kit (Thermo Fisher Scientific, Scoresby, Australia). DNA was extracted from DNase-treated EVs and from the remaining plasma (2–4 mL) using the QIAamp Circulating Nucleic Acid Kit (Qiagen, Chadstone, Australia). DNA was extracted from 10-micron sections of FFPE tissue with the GeneRead DNA FFPE Kit (Qiagen). All protocols were performed as per the manufacturer’s instructions. DNA was quantified with both the Qubit dsDNA HS Assay (Thermo Fisher Scientific) and the Agilent High Sensitivity DNA Assay (Agilent Technologies, Mulgrave, Australia) and stored at −20 °C until further use.

### 2.7. Low-Pass Whole Genome Sequencing

DNA was sheared via sonication using the Covaris S2 Focused-ultrasonicator (Covaris, Bankstown, Australia) for 40 s with duty cycles of 10,200 cycles per burst and an intensity of 5. Library preparation was performed with the NEBNext Ultra II DNA Library Preparation Kit (New England Biolabs, Ipswich, MA, USA) as per the manufacturer’s instructions. Library fragment size distribution was determined with the Agilent 4200 TapeStation System (Agilent Technologies). Libraries were normalised, pooled and diluted to a final concentration of 1.8 pM and sequenced with 75 base pair (bp) paired-end reads on the Illumina NextSeq 500 platform (Illumina, Scoresby, Australia) up to a mean depth of 1×.

### 2.8. Data Analysis

Reads were aligned to hg19 with the Burrows–Wheeler Aligner-MEM software v0.7.15 following the removal of sequencing adaptors with Trim Galore software v0.4.5 (Babraham Institute, Cambridge, UK) [17]. Variant calling was performed on the GATK 3.7 haplotype caller (The Broad Institute, Cambridge, MA, USA). Metric summarisation and duplicate identification were carried out using Picard tools v2.9.0 (The Broad Institute,). QDNASeq software v1.19.0 (VU University Medical Centre, Amsterdam, The Netherlands) was used to call the copy number variations, with no matched normal samples, the baseline ploidy set at 2 and a fixed bin size of 15 kilobases (kb) [18]. A 2-tailed Student’s *t*-test was used to determine whether there was a significant difference in EV DNA vs. ctDNA yield and in the sequencing metrics of tumour DNA (tDNA), EV DNA and ctDNA samples. A *p*-value of <0.05 was considered statistically significant.

## 3. Results

### 3.1. Patient Characteristics

The subject of this study was a 61-year-old female with Stage II invasive ductal carcinoma of the triple negative subtype, diagnosed in November 2009. The tumour was surgically resected in April 2010 and the patient was treated with adjuvant chemotherapy (4 cycles of docetaxel and cyclophosphamide) and radiotherapy. Four years later (June 2014), a chest wall lump was biopsied and shown to be metastatic breast cancer; capecitabine was then commenced. In January 2015, a computed tomography (CT) scan demonstrated mediastinal progression and oral vinorelbine was prescribed but did not result in tumour reduction. By March 2015, the mediastinal mass had enlarged and mediastinal radiotherapy and gemcitabine treatment commenced, leading to a durable response. The patient consented to the study in March 2016, whilst in remission, and blood samples were then collected consecutively over a period of 7 months from March to September 2016. Disease progression was noted in December 2016, with posterior cerebellar metastasis, and—despite surgical excision—the patient died weeks later.

### 3.2. EV DNA vs. ctDNA Yield

EVs within the range of 30–600 nm (mean = 200 ± 30 nm) were identified, indicating that samples contained an enrichment of small EVs (see Figure 1a). EVs were positive for CD9, a common protein present in small EVs (Figure 1a). The yields and size distributions of ctDNA (mean = 4.0, range = 3.7–4.6 ng/mL plasma and 172 bp, respectively; see Figure 1b,c) were consistent with those previously reported [19]. We observed significantly lower amounts of EV DNA (mean = 0.8, range = 0.6–1.1 ng/mL plasma; *p* = 0.02; see Figure 1b) and a dramatically different size profile with a uniform distribution of fragment sizes ranging from <100 to >10,000 bp (Figure 1c). This reflects the EV DNA size distribution reported in previous studies [7,10].

### 3.3. Copy Number Analysis

Copy number analysis revealed that all samples covered each autosome in an unbiased manner. The number of detectable variants in patient tumour DNA increased in each subsequent sample, with 65 variants detected in tDNA 1, 75 in tDNA 2 and 82 in tDNA 3 (see Figure 2). This was also reflected in the circulating DNA samples, with 7 variants identified in ctDNA from Plasma 1, 23 in ctDNA Plasma 2 and 48 in ctDNA Plasma 3. EV DNA Plasma 1 and Plasma 2 samples did not have any detectable variants, while Plasma 3 EV DNA contained 13 CNVs. There were significantly more CNVs detected in tDNA samples than in ctDNA or EV DNA samples (*p* = 0.04 and *p* < 0.01, respectively). Variants ranged in size from 0.3 to 106.5 megabases (Mb; mean 20.0 Mb) and were distributed randomly throughout the genome (i.e., they were not concentrated in particular chromosomes).

There were 52 individual regions of copy number variation, which were detected in all tumour DNA samples, indicating that these were likely early ‘driver’ events (those required for tumour growth and survival); these were present throughout patient treatment (see Appendix A for a full list of copy number changes and associated genes detected in all samples). Of these variants, 36 (69%) were detected in at least one sample of ctDNA and 13 (25%) were detected in EV DNA Plasma 3. There were two variants identified in ctDNA that were not in the tDNA or EV DNA. All 13 variants in EV DNA were identified in the tumoural DNA, but two were not found in the ctDNA.

## 4. Discussion

This case study has demonstrated that LP-WGS can identify CNVs in EV DNA as well as in ctDNA, the first such observation for breast cancer. Importantly, several of these CNVs have potential implications for prognosis and treatment. For example, patients with both germline and somatic loss of heterozygosity at the *BRCA1* locus (17q21) have been shown to have improved responsiveness to platinum-based chemotherapies and sensitivity to poly-ADP-ribose polymerase inhibitors, whilst also demonstrating selective resistance to taxanes [3,20,21]. The number of detectable CNVs in patient ctDNA and EV DNA increased over the time course and natural history of the disease, in keeping with disease progression and associated clonal evolution. Emerging CNVs were detectable within both ctDNA and EV DNA prior to clinical evidence of the cerebellar metastasis, suggesting that sequencing of circulating DNA may indicate disease progression at an earlier stage than conventional clinical monitoring tools.

While this is a novel and potentially significant finding, CNVs were only detected in one of the three EV DNA samples studied. EV DNA also had a considerably lower sensitivity for tumour driver CNVs than ctDNA (25% vs. 69%, respectively). Technical factors that may explain these differences in sensitivity include the very low yields (~5 ng total) and fragmented nature of EV DNA. Future studies may attempt to overcome these technical issues through the size selection of large EVs (>1000 nm), which have been shown to contain more tumour-derived DNA than small EVs (<200 nm) [9,22]. Removing the 12,000 g ultracentrifugation step may increase the yield of large EVs and thus EV DNA. Similarly, methods to enrich tumour-specific EV subpopulations (i.e., immunocapture) would increase the tumour DNA signal over the wild-type background and may improve sensitivity [23].

Furthermore, the copy number profiling method utilised does not include a matched normal sample to normalise the copy number [18]. While this does increase the sensitivity of the analysis by improving upon the level of background noise that would be generated by running a reference sample, it also means that the algorithm cannot completely correct for technical bias in read coverage [24]. As such, there is a higher margin for error in CNV calling, which could explain why there were no CNVs called with confidence in EV DNA Plasma 1 and Plasma 2 samples. There are many computational approaches to correct for this source of variance; however, all of them rely on a comparison between tumour and matched normal tissue and thus could not be applied in this case [25]. Follow-up studies should therefore endeavour to include patient-matched germline DNA to increase the confidence of copy number detection, as well as to enable the discrimination between inherited and somatic variants, as this may have bearing on their clinical significance.

Biological differences between the release and distribution of ctDNA and EV DNA may have also contributed to the observed disparities in the copy number profiles. Given that the main mechanisms of ctDNA and EV release are through cell death and active secretion, respectively, it is reasonable to hypothesise that these entities are released from two distinct subpopulations of cells—i.e., ctDNA is shed from apoptotic tumour cells, whereas EV DNA is actively secreted from viable neoplastic cells [26,27]. Because the purpose of this study was to compare the copy number profiles of EV DNA and ctDNA, a pre-extraction DNase treatment step was necessary to remove contaminating cell-free DNA from EV DNA samples. However, future studies may exclude this step in order to capture all DNA within EV preparations and maximize the likelihood of detecting a tumour-specific change.

Discordance between plasma and tissue can be attributed to genetic heterogeneity within and between tumours, as well as temporal disparities in sample collection [19]. In this study, plasma was collected from the patient several years after the resection of the primary tumour, during which time she received multiple lines of chemotherapy. As a result, it is unsurprising that there were some variants detected in the primary tumour that were not detected in the plasma, and vice versa. In addition, rates of ctDNA shedding can impact the representation of molecular drivers within the plasma [28]. This disparity can be exacerbated by the release of cell-free DNA from non-malignant cells (such as leucocytes), which dilutes the tumour fraction within the blood and decreases the sensitivity for the detection of rarer CNVs [29]. In this study, plasma processing was performed within two hours of collection to reduce the possibility of contamination via leucocyte lysis. Further optimisation of plasma collection and processing (i.e., using blood collection tubes with specialised preservatives such as Streck Cell-Free DNA BCT^®^ tubes) may lead to improvements in CNV detection.

## 5. Conclusions

This case study is significant as it demonstrates that CNVs reflecting the tumour genomic profile can be detected in both the ctDNA and EV DNA of a breast cancer patient. However, while studies in other cancers suggest that EV DNA is a viable alternative to ctDNA for genetic profiling and disease monitoring, we show that the sensitivity of EV DNA for CNV detection is lower than that of ctDNA. This may be due to the quality and quantity of EV DNA as well as biological factors, such as the cell of origin. As such, further studies of EV DNA in breast cancer are required before any comment can be made regarding the clinical applicability of this approach.

## Figures and Tables

**Figure 1 biomedicines-09-00014-f001:**
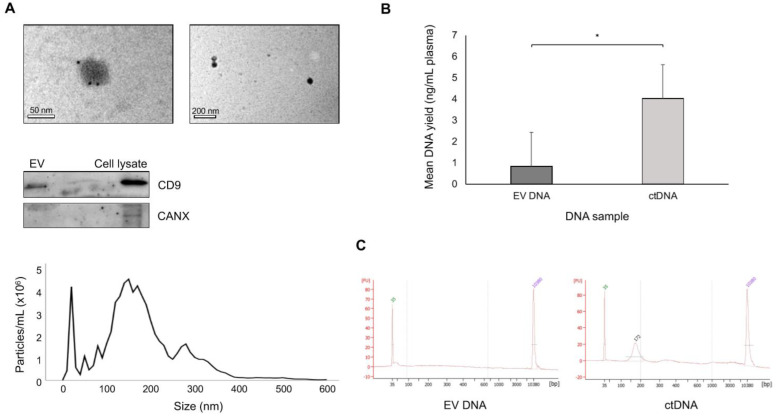
Validation of extracellular vesicle (EV) isolation and DNA extraction from plasma. (**A**) Electron micrographs (top) demonstrate successful isolation of particles approximately 50 nm in diameter with ultracentrifugation (left panel: 30,000× magnification of single EV; right panel: 8000× magnification of multiple EVs). Particles were positive for CD9 (black dots), as determined by staining of samples with gold-labelled antibodies specific for the CD9 protein. Western blotting of samples isolated with ultracentrifugation (EV; middle) confirmed the presence of CD9, as well as negativity for calnexin (CANX). A sample of cell lysate from breast cancer cell line MDA-MB-231 was used as a positive control. Nanoparticle tracking analysis (bottom) determined that particles isolated fell within the expected size range of 30–600 nm, with peaks at 30, 150 and 300 nm. (**B**) Mean ctDNA and EV DNA yields (ng/mL plasma) for patient A3 (* = *p*-value of 0.02). (**C**) Representative electropherograms of ctDNA (left) and EV DNA (right) showing the fragment size distribution profiles (bp) in relative fluorescence units (FU). Lower marker = 35 bp; upper marker = 10,380 bp. Dotted lines define regions of interest. Green lines indicate peak size region.

**Figure 2 biomedicines-09-00014-f002:**
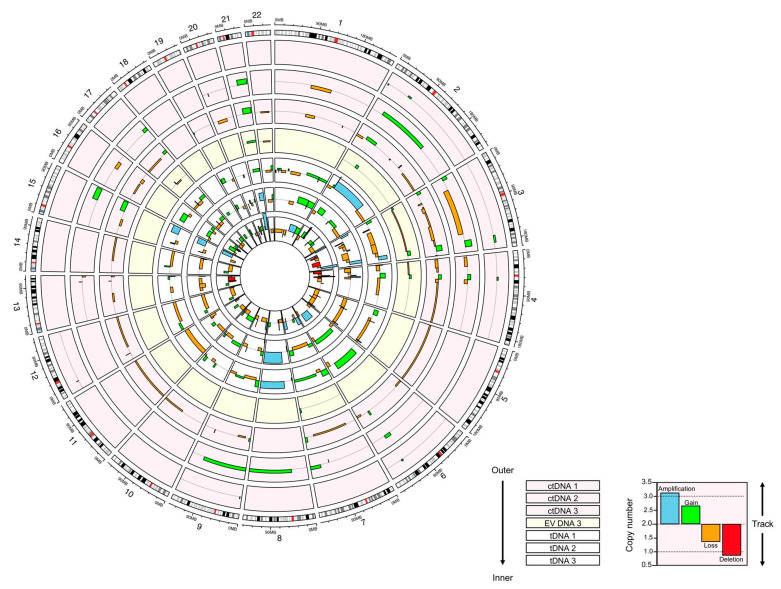
Circos plot comparing copy number alterations detected in ctDNA, EV DNA and tDNA samples. Each track illustrates the copy number across the genome in individual samples. Note that EV DNA samples 1 and 2 did not contain detectable copy number variants (CNVs) and thus are not depicted. Tracks are divided into sections according to each autosome, arranged clockwise from 1 to 22. The vertical scale of the tracks indicates the copy number at that region in the genome. The scale is consistent across all tracks. The grey line in the middle of each track represents a normal copy number of 2.0. CNVs are depicted by boxes coloured according to the type of alteration (see key). The horizontal length of the box indicates the size of the region of altered copy numbers in megabases (scale on outer ideogram).

## Data Availability

The data presented in this study are available in Appendix A.

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
