# Peer review of "Comparison of Circulating Tumour DNA and Extracellular Vesicle DNA by Low-Pass Whole-Genome Sequencing Reveals Molecular Drivers of Disease in a Breast Cancer Patient"

_biomedicines, 2020, doi:10.3390/biomedicines9010014_

Round 1
Reviewer 1 Report
The manuscript deals with the highly topical issue of liquid biopsy, which has considerable potential to improve the management of cancer patients. The authors compared the presence of CNV in tumor samples from breast cancer patients with their presence in ctDNA and EV DNA. Although such a comparison yields interesting results in this quite challenging area of research, the quality of the manuscript should be improved.
The material and methods section lacks information on the time sequence of sampling. The quality of the images is quite low (perhaps only for evaluation purposes), but the individual images (e.g. Fig.1c, a legend to Fig.2, etc.) are small and unreadable, the resolution is low. While lowercase letters are used in the caption to Figure 1, uppercase letters are used in the figure itself. The western blot image is small and of poor quality. It would be clearer if the EV DNA and ctDNA were shown in the same order in Figures 1b and 1c. In the legend to Figure 1, it is written that: “A sample of cell lysate from breast cancer cell line MDA-MB-231 was used as a positive control.” However, it is not mentioned in the Material and methods section.
Taking into account rapid progress in this area of research, the references in the manuscript are obsolete, the most recent one from 2019, the others by 2018 at the latest.
Author Response
Thank you for your review of our manuscript. Please see the attachment for responses to your comments.

Reviewer 2 Report
Ruhen et al. used low-pass whole genome sequencing to identify the copy number variants in both ctDNA and EV DNA from breast cancer patients. They found that ctDNA has higher sensitivity for the detection and it is more useful for monitoring of breast cancer. This manuscript is interesting and well written. Only minor modifications are required for publication.
Minor comments
- Figure 1A. It was not clear the explanation of Figure 1A upper left panel. What are difference between left and right pictures? The left picture has three CD9 signals, but the right panel has only one dot?
- Figure 1C. The figure is very hard to see. The authors should re-plot them to see clearer.
- Figure 2. I could not see the type of alteration in the bottom panel.
- Is the copy number alteration accumulated in particular position of chromosomes? Can the authors comments on the positions and/or relationship with chromatin structure?
Reference format should be consistent. E.g. ref1 and 3 are different from others.
Author Response
Thank you for your review of our manuscript. Please see the attachment for our response to your comments.
